# Investigation on Direct Shear and Energy Dissipation Characteristics of Iron Tailings Powder Reinforced by Polypropylene Fiber

**Ping Jiang, Shaowei Lv, Yue Wang, Na Li *** and **Wei Wang ***

School of Civil Engineering, Shaoxing University, Shaoxing 312000, China; jiangping@usx.edu.cn (P.J.); usxluishaowei@163.com (S.L.); wangyueusx@163.com (Y.W.)
* Correspondence: lina@usx.edu.cn (N.L.); wellswang@usx.edu.cn (W.W.)

**Abstract:** Resource utilization of iron tailings powder is an effective measure to reduce the dam-break risk of an iron tailings reservoir. Adding polypropylene fiber to iron tailings powder can improve its shear performance. Direct shear tests were carried out on reinforced iron tailings powder with polypropylene fiber with dosages of 0%, 0.25%, 0.5%, 0.75%, and 1%, respectively. The normal stresses during the tests were 100, 200, 300, and 400 kPa, respectively. The test results show that with the increase of polypropylene fiber dosage, the cohesive force of iron tailings powder firstly increases and then decreases gradually, and the internal friction angle firstly decreases and then increases. The back propagation (BP)neural network was used to fit the shear force ($F$) and shear displacement ($s$) of the test to obtain the $F$-$s$ function relationship that satisfies the accuracy. Based on the energy dissipation theory, the direct shear energy dissipation of polypropylene-fiber-reinforced iron tailings powder was calculated. The mathematical model of energy dissipation of fiber interfacial failure was derived by the fiber distribution model. The interfacial strength parameters of polypropylene fiber were calculated based on the direct shear test data and the mathematical model of fiber interfacial energy dissipation. The test results show that the addition of polypropylene fiber from the perspective of energy dissipation can improve the shear properties of iron tailings powder.

**Keywords:** iron tailings powder; polypropylene fiber; shear characteristics; energy dissipation; interfacial strength parameter

## 1. Introduction

Iron tailings are one of the products of iron ore sorting operations. The stacking of tailings not only affects the surrounding environment, but also seriously threatens the lives and property security of the people in the lower reaches of the reservoir area. Therefore, how to use those iron tailings to turn waste into treasure is an effective means to reduce the dam-break risk of an iron tailings reservoir. Much research has been carried out on the resource utilization of iron tailings powder, including concrete fine aggregate [1], sewage treatment [2], filling material of composite material [3], recovered iron ore [4], and as a road material [5]. Lizhu Iron Mine is located in Shaoxing city, Zhejiang province, China. It produces about 200,000 tons of dewatered tailings every year and accumulates more than 20 million tons of tailings in the existing tailings dam yard. The iron tailings mainly contain tailings with a particle diameter of less than 0.075 mm, which are cohesionless, non physicochemically active, and difficult to utilize. Certain means shall be taken to improve its resource utilization.

Adding fiber to soil can improve the mechanical properties of soil through the tensile properties of fiber and the interfacial strength between fiber and soil [6–9]. Consoli et al., Diambra et al., and Li and Senetakis studied the mechanical properties and test methods of fiber-reinforced sand, such as

interfacial characteristics, constitutive model, shear modulus, and damping ratio, through triaxial tests and a resonance columns test [10–13]. Diambra proposed a constitutive model of fiber-reinforced clay and concluded that fiber reinforcement can effectively improve the undrained shear strength of soil [14]. Diab conducted the comparative study on two kinds of fiber-reinforced clay sample production methods of "impact" and "kneading" through an unconsolidated undrained triaxial test [15]. Soltani et al. studied the effects of fiber width, fiber dosage, and fiber length on the swelling properties of expansive soils [16]. Taking the effects of fiber dosage, fiber length, relative compactness, and confining pressure into consideration, Chegenizadeh et al. carried out a series of cyclic triaxial tests on unreinforced and reinforced specimens to study the effect of bulk continuous fibers (BCF) on the liquefaction resistance of low plasticity silt [17]. It can be seen from the above research results that fiber materials are widely used in sand, clay, expansive soil, and silt, which can improve the mechanical properties of the soil to some extent. Therefore, fiber can be added to iron tailings powder in order to improve the mechanical properties.

Energy dissipation is a theoretical method which is often used to analyze the material failure characteristics, such as the calculation of slope safety factor, the constitutive relationship of geomaterials, the interfacial analysis of soil and structure, and the mechanical properties of soil and cement soil [18–28]. Ibraim et al. explored the energy consumption of unreinforced and reinforced sand samples based on experimental and discrete element modeling (DEM) procedures [29]. On the basis of energy, Amini and Noorzad investigated the effect of fiber dosage, fiber length, confining pressure, and relative density on the cyclic shear resistance of fiber-reinforced sand [30]. Since energy is scalar, the shear energy dissipation of polypropylene-fiber-reinforced iron tailings powder can be regarded as the sum of the fiber interfacial energy dissipation and the direct shear energy dissipation of iron tailings powder.

Shear performance is the most basic characteristic of geomaterials. Based on the energy dissipation theory, and referring to the research results of fiber-reinforced soil, the shear properties and interfacial strength parameters of fiber-reinforced iron tailings powder are studied. In this study, the shear properties of polypropylene-fiber-reinforced iron tailings powder were investigated by a direct shear test. The energy dissipation during the direct shearing process was analyzed to further derive the fiber interfacial strength parameters. The shear performance mechanism of polypropylene-fiber-reinforced iron tailings powder was analyzed to provide reference for the resource utilization of iron tailings powder.

## 2. Experimental Material

The iron tailings powder used in this test comes from Lizhu Iron Mine in Zhejiang province, as shown in Figure 1. The specific gravity and specific surface area of iron tailings powder were 3.06 and 379 $m^2$/kg, respectively, through physical property tests.

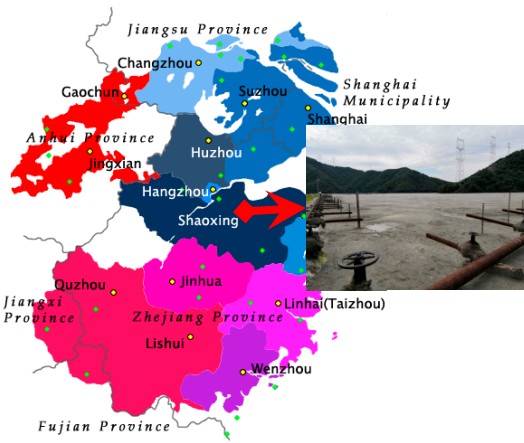

**Figure 1.** Zhejiang Lizhu Iron tailings mine.

Particle size analysis of iron tailings powder was conducted by a Mastersizer 2000 laser particle size analyzer to obtain the particle content of less than 45, 45–75, 75–100, 100–150, and more than 150 μm. The test results are shown in Table 1.

**Table 1.** Particle size analysis results of iron tailings powder.

| Particle Diameter (μm) | <45 | 45–75 | 75–100 | 100–150 | >150 |
|---|---|---|---|---|---|
| Content (%) | 69.57 | 8.76 | 5.18 | 6.98 | 9.51 |

The fiber material used in the test was polypropylene fiber as shown in Figure 2. The fiber form is monofilament bunchiness, the tensile strength and elastic modulus were 260 and 3800 MPa, the specific gravity was 0.91, and the length and diameter were 6 and 0.023 mm, respectively.

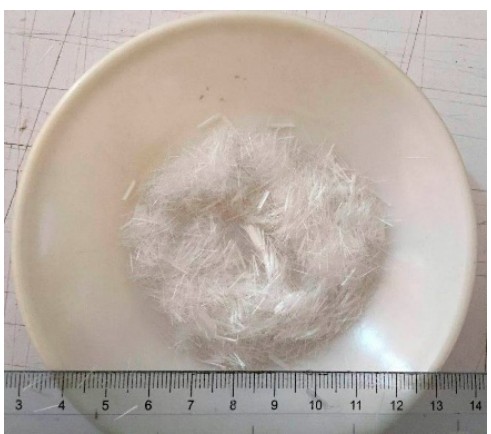

**Figure 2.** Polypropylene fiber.

## 3. Direct Shear Test

Since Lizhu iron tailings powder is mainly particles with a diameter of less than 45 μm, it can be reinforced by polypropylene fibers. The shear properties of the material are the most basic mechanical indexes. The shear properties of polypropylene-fiber-reinforced iron tailings powder were studied through a direct shear test.

### 3.1. Sample Preparation and Testing

The dosage of polypropylene fiber used in the test was the percentage of fiber relative to the dry mass of iron tailings powder. According to the research results of fiber-reinforced sand, clay, expansive soil, and silt [10–17], the polypropylene fiber dosages selected here were 0%, 0.25%, 0.5%, 0.75%, and 1%, respectively. In the sample preparation process, 17% water was first added to iron tailings powder and stirred for 3 min, and then let stand for 24 h. Then, the fibers were sprinkled into iron tailings powder and stirred for 3 min. The mixture is shown in Figure 3. The mixture was placed into a ring cutter of 61.8 mm in diameter and 20 mm in height, and a layer of Vaseline was applied inside the ring cutter before loading [31]. The sample was formed by compaction, and the upper and lower surfaces of the sample were flattened [32]. The formed sample is shown in Figure 4. The mold was released after 2 h, and the weight of each sample was weighed and controlled at 146 ± 2 g. The sample was packed in cling film and placed in a natural environment for 24 h, and then subjected to a direct shear test.

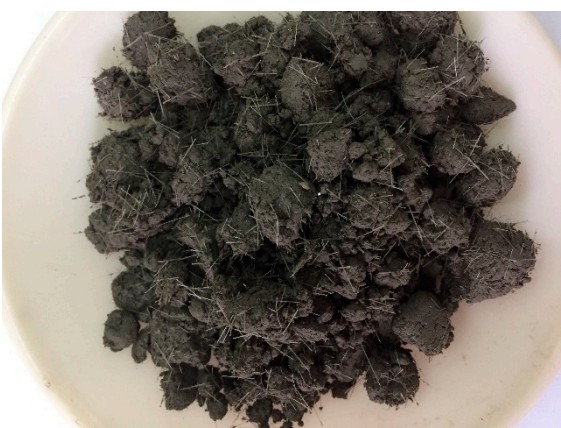

**Figure 3.** Mixture of polypropylene fiber and iron tailings powder.

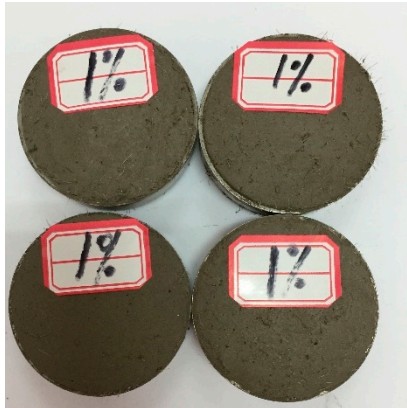

**Figure 4.** Direct shear test specimens of polypropylene-fiber-reinforced iron tailings powder with 1% fiber dosage.

The full-automated quadruple direct shear apparatus produced by Nanjing TKA Technology Co., Ltd. was used in this test. It can perform direct shear tests under four normal stresses simultaneously, thereby improving the test efficiency. The normal stresses used in this test were 100, 200, 300, and 400 kPa, respectively, and the shear rate was 1 mm/min. Six shear forces (*F*) and the corresponding shear displacement (*s*) were recorded every second during the test, and the sample failure shear surface is shown in Figure 5.

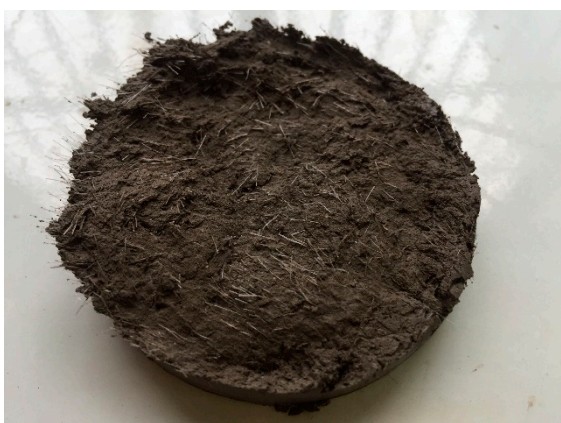

**Figure 5.** Shear surface of polypropylene-fiber-reinforced iron tailings powder with 1% fiber dosage.

## 3.2. Test Results and Analysis

According to the test data, the relationship between shear force (*F*) and shear displacement (*s*) of polypropylene-fiber-reinforced iron tailings powder can be plotted, as shown in Figure 6. In Figure 6, (a)–(e) correspond to the *F-s* curve of the sample under the normal stresses of 100, 200, 300, and 400 kPa when the polypropylene fiber dosage is 0%–1%.

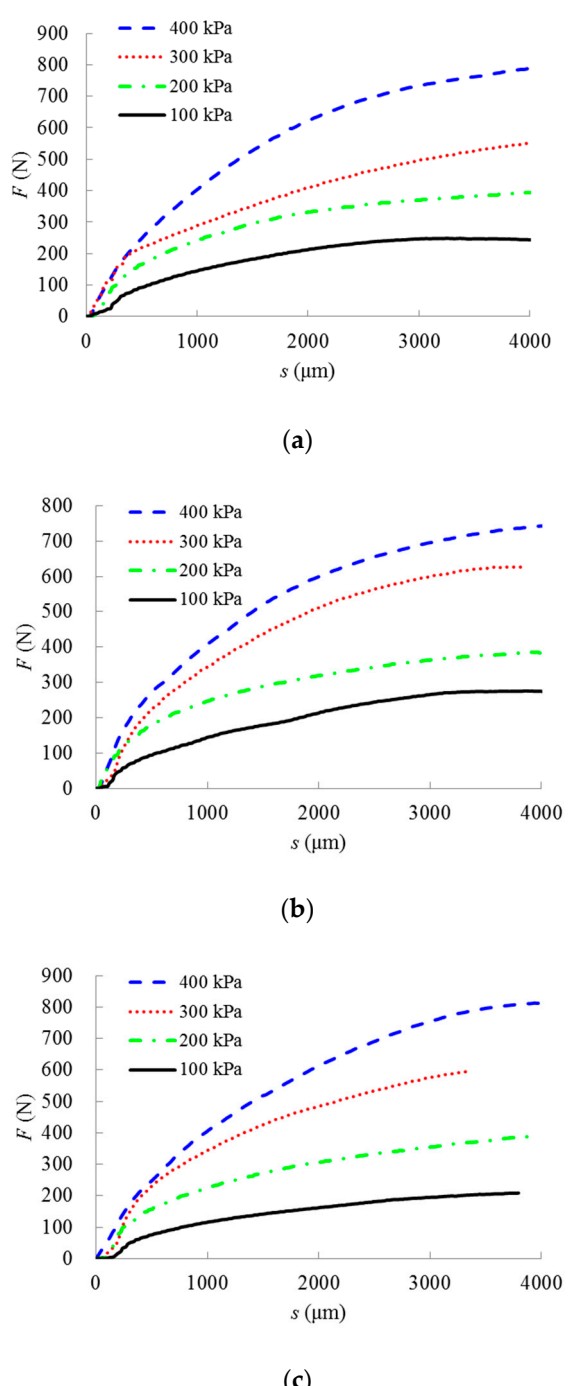

**Figure 6.** *Cont.*

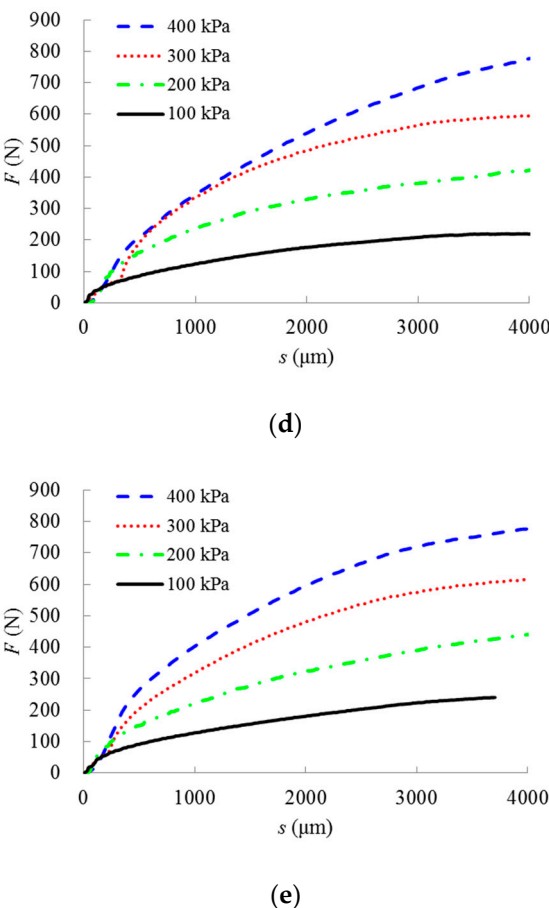

**Figure 6.** *F-s* curve with different polypropylene fiber dosages. (**a**) 0% polypropylene fiber dosage [33] (**b**) 0.25% polypropylene fiber dosage; (**c**) 0.5% polypropylene fiber dosage; (**d**) 0.75% polypropylene fiber dosage; (**e**) 1% polypropylene fiber dosage.

According to the F-s curve, the shear force corresponding to the 4 mm shear displacement is the peak force $F_{max}$ [34]. The peak shear strength $\tau_{max}$ is calculated according to Equation (1). According to the Mohr–Coulomb principle (Equation (2)), the cohesive force c and the internal friction angle $\varphi$ of the reinforced iron tailings powder with different polypropylene fiber dosages can be calculated and plotted in Figure 7.

$$\tau_{max} = \frac{F_{max}}{A} \tag{1}$$

$$\tau = c + \sigma \tan \varphi \tag{2}$$

It can be seen from Figure 7 that with the addition of polypropylene fibers, the cohesive force firstly increases and then gradually decreases. When the dosage was increased from 0% to 0.25%, the cohesive force increased from 12.3 to 31.9 kPa. When the dosage was increased from 0.25% to 1%, the cohesive force gradually decreased from 31.9 to 9 kPa, which was lower than that of iron tailings powder with no fiber dosage. When the fiber dosage was increased from 0% to 0.25%, a small amount of fiber filled the pores between the iron tailings powder, taking advantage of the spatial constraint effect, so as to increase the cohesive force of fiber-reinforced iron tailings powder. When the fiber dosage was increased from 0% to 0.25%, the agglomeration effect of fiber was increased due to the increase of fiber. The cohesive force of fiber-reinforced iron tailings powder was decreased on account of the spatial constraint effect finding it hard to proceed.

With the increase of fiber dosage, the internal friction angle of polypropylene-fiber-reinforced iron tailings powder firstly decreases and then increases. When the fiber dosage was increased from 0% to 0.25%, the internal friction angle was reduced from 31.7 to 29.3°. When the fiber dosage was increased

from 0.25% to 1%, the internal friction angle increased from 29.3 to 35.3°. When the fiber dosage was increased from 0% to 0.25%, the fiber was dispersed among the iron tailings powder. A small amount of fiber cannot produce an obvious friction effect, and the internal friction angle of iron tailings was decreased due to the particle spacing of iron tailings being increased. When the fiber dosage was increased from 0.25% to 1%, the combination form between fiber and iron tailings was diversified due to the fiber density in the same volume being increased, and the angle of internal friction was large due to the strengthening of the friction effect.

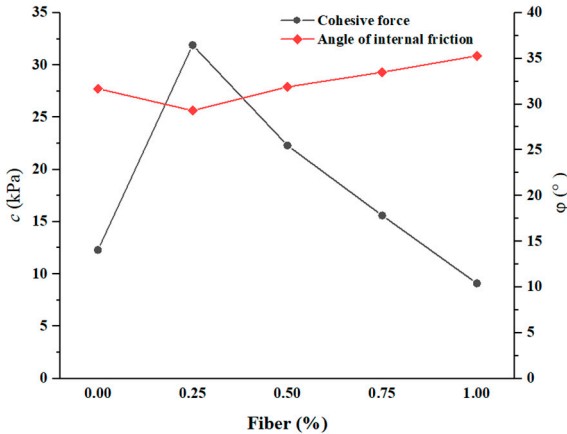

**Figure 7.** Cohesion and internal friction angle of fiber-reinforced iron tailings powder with different polypropylene fiber dosages.

It can be concluded from the direct shear test results that when the fiber dosage is between 0% and 0.25%, the cohesive force increases, the internal friction angle decreases; when the fiber dosage is between 0.25% and 1%, the cohesive force decreases, and the internal friction angle increases. Therefore, it is difficult to analyze the reinforcement effect of polypropylene fiber on iron tailings powder using the variation law of cohesive force and internal friction angle. Other parameters are needed to describe the reinforcement effect of polypropylene fiber.

## 4. Shear Energy Dissipation of Fiber-Reinforced Iron Tailings Powder

The direct shear failure process is actually the energy dissipation process. According to the first law of thermodynamics, the shear force can be used to characterize the energy dissipation. Since the discrete data points of $F$ and $s$ were obtained during the test, it was difficult to calculate the energy dissipation. Therefore, it is necessary to establish a continuous $F$-$s$ function relationship and calculate the direct shear energy dissipation of polypropylene-fiber-reinforced iron tailings powder by means of a definite integral. A BP neural network algorithm is a commonly used calculation method for data fitting. An appropriate neural network structure can simulate any nonlinear function relationship [35–38]. Therefore, a BP neural network algorithm was used to fit the $F$-$s$ function relationship and calculate the direct shear energy dissipation of polypropylene-fiber-reinforced iron tailings powder, thus analyzing the fiber reinforcement effect.

### 4.1. F-s Curve Fitting Based on a BP Neural Network

In this paper, a BP neural network algorithm was used to fit the $F$-$s$ curve. The network structure contains one input layer node, two hidden layer nodes, and one output layer node. The transfer functions of the hidden layer and output layer are shown in Equations (3) and (4), and the shear displacement $s$ obtained by the direct shear test represents the input variable, and the shear force $F$ represents the output variable [33,39].

$$f(v) = \frac{1}{e^{-v} + 1} \tag{3}$$

$$f(v) = v \tag{4}$$

In order to ensure the fitting accuracy, the input variables and output variables need to be normalized [38,39]. The fitting function of *F-s* is obtained as

$$F(s) = \frac{a_1}{e^{-a_2 s - a_3} + 1} + \frac{a_4}{e^{-a_5 s - a_6} + 1} + a_7, \tag{5}$$

where $a_1 = \frac{F_{max}\omega_{21}}{2}$, $a_2 = \frac{2\omega_{11}}{s_{max}}$, $a_3 = b_{11} - \omega_{11}$, $a_4 = \frac{F_{max}\omega_{22}}{2}$, $a_5 = \frac{2\omega_{12}}{s_{max}}$, $a_6 = b_{12} - \omega_{12}$, $a_7 = \frac{F_{max}}{2}(b_2 + 1)$, among them, $\omega_{11}$, $\omega_{12}$, $b_{11}$, $b_{12}$, $\omega_{21}$, $\omega_{22}$, and $b_2$ are BP neural network calculation parameters, the specific meaning of which can be found in reference [39]. Table 2 shows the calculation results of BP neural network parameters and the average fitting error. Figure 8a,b shows the comparison of the F-s fitting curve and the test curve of polypropylene-reinforced iron tailings powder under normal stresses of 100, 200, 300, and 400 kPa with 0.5% and 1% fiber dosages.

**Table 2.** BP neural network fitting results.

| Fiber Dosage (%) | Normal Stress (kPa) | $a_1$ | $a_2$ | $a_3$ | $a_4$ | $a_5$ | $a_6$ | $a_7$ | Average Error (N) |
|---|---|---|---|---|---|---|---|---|---|
| 0 | 100 | −49.93 | 0.0023 | −9.71 | 727,324 | 0.00077 | 7.87 | −727,051 | 0.03 |
| | 200 | −19.77 | 0.0055 | −29.97 | −4,557,279 | −0.00091 | −9.36 | 399 | 0.06 |
| | 300 | −14.06 | 0.0033 | −17.77 | 317,668 | 0.00037 | 6.33 | −317,062 | 0.05 |
| | 400 | 24.61 | −0.0062 | 34.9 | −38,062 | −0.00067 | −3.79 | 823 | 0.02 |
| 0.25 | 100 | 111.6 | 0.0022 | −4.46 | 189,740 | 0.0017 | 6.99 | −189,573 | 0.05 |
| | 200 | −8044 | −0.00056 | −3.26 | 668 | 0.0053 | 1.41 | −248 | 0.02 |
| | 300 | 1230 | 0.001 | −0.03 | −94 | −0.02 | 6.82 | −514 | 0.07 |
| | 400 | −1634 | −0.00087 | −0.37 | −125 | −0.016 | 2.3 | 775 | 0.02 |
| 0.5 | 100 | −28,952 | −0.0005 | −4.92 | 39.95 | 0.021 | −5.78 | 202 | 0.38 |
| | 200 | 243,556 | 0.00052 | 6.54 | −93.45 | −0.0187 | 3.67 | −243,126 | 0.08 |
| | 300 | 112,578 | 0.0005 | 5.23 | −116 | −0.018 | 4.83 | −111,876 | 0.13 |
| | 400 | −262 | −0.0018 | 3.98 | −17,714 | −0.0011 | −3.4 | 827 | 0.01 |
| 0.75 | 100 | −270 | −0.001 | 0.56 | 338,718 | 0.0068 | 8.73 | −338,489 | 0.01 |
| | 200 | −169,731 | −0.00056 | −6.1 | 80.82 | 0.019 | −3.74 | 375 | 0.13 |
| | 300 | 903 | 0.001 | −0.24 | −101.5 | −0.016 | 6.99 | −294 | 0.01 |
| | 400 | −1737 | −0.0006 | −0.137 | −111 | −0.0117 | 3.08 | 903 | 0.04 |
| 1 | 100 | 425 | 0.0006 | −0.09 | 295,144 | 0.006 | 8.4 | −295,283 | 0.01 |
| | 200 | −81,430 | −0.00038 | −5.16 | 99.53 | 0.0125 | −1.35 | 438 | 0.03 |
| | 300 | 1014 | 0.0009 | −0.18 | −92.58 | −0.0151 | 4.52 | −365 | 0.01 |
| | 400 | −1500 | −0.00076 | −0.114 | 148 | 0.0146 | −3.99 | 693 | 0.02 |

It can be seen from Table 2 and Figure 8 that the BP neural network can better fit the *F-s* function relationship with a maximum fitting error of 0.4 N. Therefore, the BP neural network algorithm can be used to fit the *F-s* function relationship in the direct shear test.

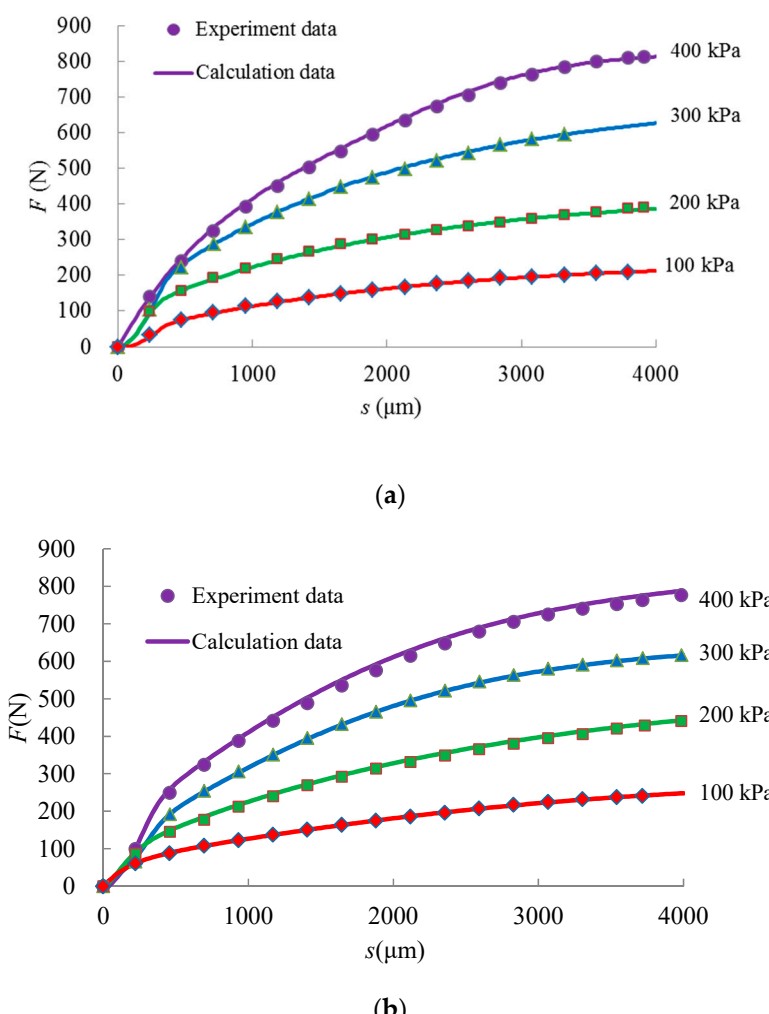

**Figure 8.** *F–s* fitting curve of polypropylene-reinforced iron tailings powder. (**a**) 0.5% fiber; (**b**) 1% fiber.

### 4.2. Calculation of Shear Energy Dissipation of Fiber-Reinforced Iron Tailings Powder

According to the *F-s* fitting function, the direct shear energy dissipation of fiber-reinforced iron tailings powder with different normal stresses can be obtained by a definite integral. The calculation formula (Equation (6)) is [39]

$$W = \int_{0}^{s_{max}} F(s)ds, \tag{6}$$

where, according to Section 4.7 of the Highway Geotechnical Test Specification, $s_{max}$ can be taken as 4 mm [34]. According to Equation (6), the direct shear energy dissipation of fiber-reinforced iron tailings powder under different normal stresses can be obtained. The calculation results are shown in Table 3.

**Table 3.** Shear energy dissipation of iron tailing powder with different fiber doses under different normal stresses (J).

| | Normal Stress (kPa) | | | |
|---|---|---|---|---|
| Fiber (%) | 100 | 200 | 300 | 400 |
| 0 | 0.74 | 1.18 | 1.24 | 2.2 |
| 0.25 | 1.08 | 1.52 | 1.99 | 2.96 |
| 0.5 | 0.83 | 1.51 | 2.38 | 2.36 |
| 0.75 | 0.87 | 1.68 | 2.36 | 3.24 |
| 1 | 0.95 | 1.76 | 2.41 | 3.13 |

The data in Table 3 is plotted as shown in Figure 9. The direct shear energy dissipation increases linearly with the increase of normal stress. Under the same normal stress, the direct shear energy dissipation of fiber-reinforced iron tailings powder is greater than that of iron tailings powder.

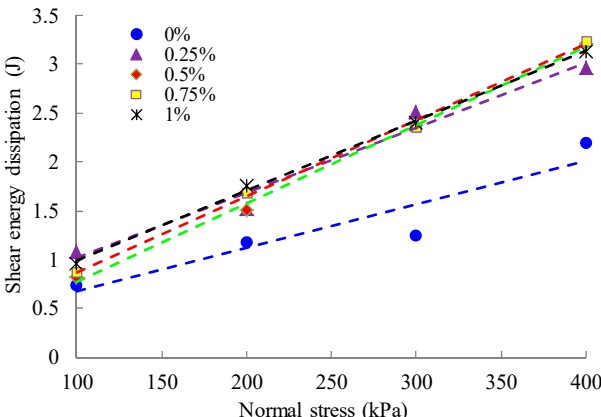

**Figure 9.** Shear energy dissipation of fiber-reinforced iron tailing powder under different normal stresses.

According to Table 3 and Figure 9, there is a linear relationship between direct shear energy dissipation, normal stress, and fiber content. The effect of fiber dosage on the direct shear energy dissipation of iron tailings powder is different. When the normal stress is 100 kPa and the fiber dosage is 0.25%, it has the maximum direct shear energy dissipation. When the normal stress is 200 and 300 kPa and the fiber dosage is 1%, it has the maximum direct shear energy dissipation. When the normal stress is 400 kPa and the fiber dosage is 0.75%, it has the maximum direct shear energy dissipation. Therefore, it is necessary to comprehensively analyze the influence of normal stress and fiber dosage on the direct shear energy dissipation of polypropylene-fiber-reinforced iron tailings powder.

## 5. Interfacial Strength Parameters of Fiber-Reinforced Iron Tailings Powder

It can be seen from the direct shear test results that polypropylene fiber has a certain effect on the shear properties of iron tailings powder and can increase energy dissipation during the direct shear process. The interfacial strength of polypropylene fiber and iron tailings powder is an important reason for the increase of direct shear energy dissipation. Therefore, it is necessary to analyze the direct shear interfacial strength parameters of polypropylene fiber and iron tailings based on the variation characteristics of energy dissipation to reveal the shearing mechanism of polypropylene-fiber-reinforced iron tailings powder.

### 5.1. Fiber Interfacial Energy Dissipation Calculation

The fiber interfacial strength can be expressed as [14]

$$\sigma_f^L = 2\frac{l_f}{d_f}(a_{sf} + p * \tan \delta_{sf}), \tag{7}$$

where $l_f$ represents the fiber length, $d_f$ represents the fiber diameter, and $a_{sf}$ and $\delta_{sf}$ represent the fiber interfacial strength parameters.

The volume density of fibers in iron tailings powder is [10]

$$\rho(\theta) = \frac{3}{2}\frac{V_f}{V}\cos^2\theta, \tag{8}$$

where $\theta$ represents the angle between fiber and the horizontal direction, $V_f$ represents the volume of fiber in the direct shear sample, and $V$ represents the volume of the direct shear sample.

Under the action of normal stress $p$, the fiber normal stress on the $\theta$-direction is shown in Figure 10 [40].

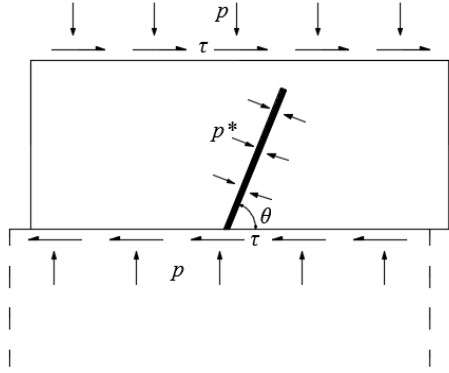

**Figure 10.** Normal stress in $\theta$-direction [40].

$$p* = p\frac{1 + \sin\varphi\sin(\varphi + 2\theta)}{\cos^2\varphi}. \tag{9}$$

In order to calculate the energy dissipation of fiber interface failure, the $\theta$-direction micro unit of fiber-reinforced soil was analyzed, as shown in Figure 11.

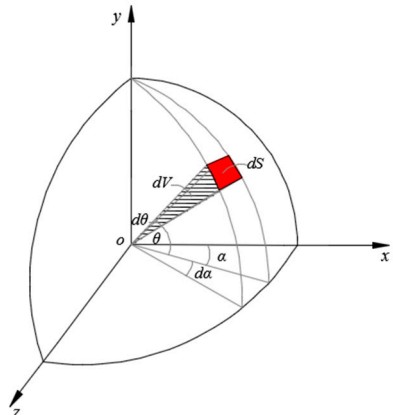

**Figure 11.** $\theta$-direction micro unit volume [14].

The volume of the $\theta$-direction micro unit is

$$dV = \int_0^{\frac{l_f}{2}} dSdl = \int_0^{\frac{l_f}{2}}\int_0^{2\pi} l^2\cos\theta d\alpha dl d\theta = \frac{\pi l_f^3\cos\theta d\theta}{12}. \tag{10}$$

The number of fibers in the micro unit is

$$m = \frac{\rho(\theta)dV}{f_v}. \tag{11}$$

In Equation (11), $f_v$ represents the volume of a single fiber:

$$f_v = \frac{\pi d_f^2 l_f}{4}. \tag{12}$$

Substituting Equations (10) and (12) into Equation (11) can obtain the number of fibers in the micro unit:

$$m = \frac{V_f l_f^2 \cos^3 \theta d\theta}{2V d_f^2}. \tag{13}$$

The energy dissipation of a single fiber interface failure is

$$W_f^1 = \frac{\pi}{4} d_f l_f^2 \sigma_f^L. \tag{14}$$

The energy required to break the fiber interface in the micro unit is

$$dW_f = m W_f^1. \tag{15}$$

Substituting Equations (7), (9), (13), and (14) into Equation (15) can obtain

$$dW_f = C_1 (a_{sf} + (C_2 + C_3 \sin(\varphi + 2\theta))\tan \delta_{sf}) \cos^3 \theta d\theta, \tag{16}$$

where $C_1 = \frac{\pi V_f l_f^5}{4 d_f^2 V}$, $C_2 = \frac{p}{\cos^2 \varphi}$, $C_3 = \frac{p \sin \varphi}{\cos^2 \varphi}$. $C_1$ is related to fiber dosage, fiber length lf and fiber diameter df, and $C_2$ and $C_3$ are related to normal stress and internal friction angle $\varphi$ of iron tailings powder.

Since fiber can only withstand the tensile force but cannot withstand the pressure, the fiber interfacial energy dissipation during the direct shear test can be expressed as

$$W_f = \int_0^{\frac{\pi}{2}} dW_f. \tag{17}$$

Substituting Equation (16) into Equation (17) can obtain the fiber interfacial energy dissipation,

$$W_f = \frac{2}{3} C_1 (a_{sf} + (C_2 + C_3 C_4)\tan \delta_{sf}), \tag{18}$$

where $C_4 = \frac{11}{10}\sin \varphi + \frac{3}{5}\cos \varphi$ is related to the internal friction angle of iron tailings powder.

The calculation results of the reinforced iron tailings powders $C_1$, $C_2$, $C_3$, and $C_4$ with different fiber dosages under different normal stresses are shown in Table 4.

**Table 4.** Calculation results of $C_1$, $C_2$, $C_3$, and $C_4$ under normal stress.

| Fiber Dosage (%) | $V_f$ (cm³) | Normal Stress $p$ (kPa) | $C_1$ (m³ × 10⁻³) | $C_2$ (kPa) | $C_3$ (kPa) | $C_4$ |
|---|---|---|---|---|---|---|
| 0.25 | 0.343 | 100 | 0.066 | 104.75 | 22.32 | 0.82 |
| | | 200 | 0.066 | 209.51 | 44.63 | 0.82 |
| | | 300 | 0.066 | 314.26 | 66.95 | 0.82 |
| | | 400 | 0.066 | 419.02 | 89.26 | 0.82 |
| 0.5 | 0.686 | 100 | 0.132 | 104.75 | 22.32 | 0.82 |
| | | 200 | 0.132 | 209.51 | 44.63 | 0.82 |
| | | 300 | 0.132 | 314.26 | 66.95 | 0.82 |
| | | 400 | 0.132 | 419.02 | 89.26 | 0.82 |
| 0.75 | 1.028 | 100 | 0.198 | 104.75 | 22.32 | 0.82 |
| | | 200 | 0.198 | 209.51 | 44.63 | 0.82 |
| | | 300 | 0.198 | 314.26 | 66.95 | 0.82 |
| | | 400 | 0.198 | 419.02 | 89.26 | 0.82 |
| 1 | 1.373 | 100 | 0.264 | 104.75 | 22.32 | 0.82 |
| | | 200 | 0.264 | 209.51 | 44.63 | 0.82 |
| | | 300 | 0.264 | 314.26 | 66.95 | 0.82 |
| | | 400 | 0.264 | 419.02 | 89.26 | 0.82 |

### 5.2. Interfacial Strength Parameters of Fiber-Reinforced Iron Tailings Powder

The direct shear energy dissipation $W_{fs}$ of fiber-reinforced iron tailings powder can be expressed as the sum of the direct shear energy dissipation $W$ of iron tailings powder and the interfacial energy dissipation $W_f$ of fiber, as shown in Equation (19).

$$W_{fs} = W_s + W_f \tag{19}$$

According to Equations (6), (18), and (19), the fiber interfacial strength parameters $a_{sf}$ and $\delta_{sf}$ can be obtained by the programming solution, and the mathematical model is shown in Equation (20).

$$\text{min} z = \sum \left| (W^i_{fs} - W^i_s) - W^i_f \right|$$
$$s.t. \begin{cases} W^i_f = \frac{2}{3} C^i_1 (a_{sf} + (C^i_2 + C^i_3 C^i_4) \tan \delta_{sf}) \\ W^i_{fs} = \int_0^{s_{max}} F^i_{fs} ds \\ W^i_s = \int_0^{s_{max}} F^i_s ds \\ a_{sf} \geq 0, \delta_{sf} \geq 0 \end{cases} \tag{20}$$

where $i$ represents the test number, $i$ = 1, 2, ..., 16. Substituting the data of Tables 2 and 3 into Equation (20), using the programming solver to obtain when $z$ is the minimum, $a_{sf}$ = 0.29 kPa, $\delta_{sf}$ = 0.57°. Substituting the calculation results of $a_{sf}$ and $\delta_{sf}$ into Equation (18) can obtain the fiber interfacial failure energy dissipation as shown in Figure 12.

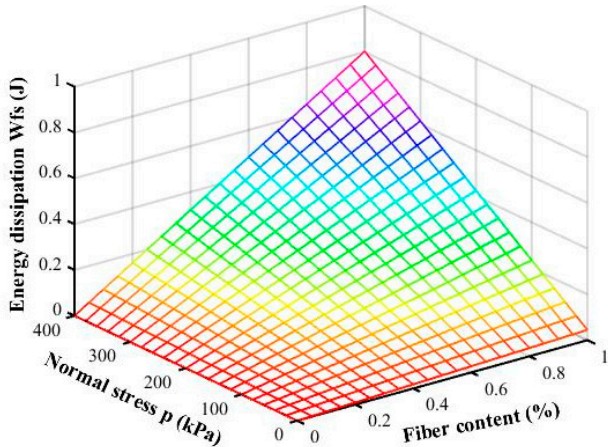

**Figure 12.** Fiber interfacial failure energy dissipation.

It can be seen from Figure 12 that under the same normal stress, as the fiber dosage increases, $W_f$ increases linearly but not obviously. In the case of the same fiber dosage, $W_f$ increases substantially and linearly with increasing normal stress. When the fiber dosage is between 0% and 1%, and the normal stress is in the range 100–400 kPa, the maximum value of $W_f$ is 0.8 J.

It can be concluded from the above analysis that the interfacial energy dissipation of polypropylene fiber and iron tailings powder increases the direct shear energy dissipation of polypropylene-fiber-reinforced iron tailings powder to a certain extent, thereby improving the shearing performance of polypropylene fiber.

## 6. Conclusions

In this paper, the shear properties and direct shear energy dissipation of polypropylene-fiber-reinforced iron tailings powder were studied by a direct shear test. Conclusions can be deduced as follows:

1. Adding polypropylene fiber can improve the cohesion of iron tailings powder, but with the increase of fiber dosage, the cohesive force gradually decreases, and the internal friction angle of iron tailings powder firstly decrease and then gradually increase.

2. A BP neural network algorithm can be used to fit the function relationship between shear force and shear displacement of polypropylene-fiber-reinforced iron tailings powder with an average error of less than 0.4 N.

3. Based on the principle of energy dissipation, the direct shear energy dissipation of polypropylene-fiber-reinforced iron tailings powder increases linearly with the increase of normal stress. Combined with the fiber distribution model, the interfacial strength parameters of polypropylene fibers and the mathematical model of fiber interfacial energy dissipation can be obtained based on the direct shear test data. Taking energy dissipation as an indicator, the addition of polypropylene fiber can improve the shear performance of iron tailings powder.

**Author Contributions:** The authors confirm contribution to the paper as follows: N.L. proposed the idea; S.L. and Y.W. conducted the tests and analyzed the data; P.J. wrote and revised the manuscript; W.W. and N.L. reviewed the results and approved the final version of the manuscript.

**Funding:** This research was funded by the National Natural Science Foundation of China (Grant number [41772311]), the Zhejiang Provincial Natural Science Foundation of China (Grant number [LY17E080016]), and the Open Research Fund of State Key Laboratory of Geomechanics and Geotechnical Engineering, Institute of Rock and Soil Mechanics, Chinese Academy of Science (Grant number [Z017013]).

**Conflicts of Interest:** The authors declare no conflict of interest.

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
