# Peer review of "Investigation on Direct Shear and Energy Dissipation Characteristics of Iron Tailings Powder Reinforced by Polypropylene Fiber"

_applsci, doi:10.3390/app9235098_

Round 1
Reviewer 1 Report
This paper presents the results of shear test on iron tailings powder reinforced with polypropylene fiber. The cohesive force and internal friction angle were measured and analyzed.
Many of the experimental observations presented in the paper require a more thorough analysis and reasoning. For example in the first paragraph on page 7, the reasons for observing the reported changes in the cohesive force should be added. The same is true about the observations regarding the internal friction angle, as reported in paragraph 2.
Also, in the first paragraph on page 10, the relation between the normal stress, the fiber dosage, and the shear energy dissipation should be justified.
The article requires a thorough English review.
Author Response
The authors sincerely thank the reviewer for taking the time and effort to review our manuscript, raise key issues and questions, and provide valuable comments and feedback.Thanks for the encouraging recommendation for the revised draft.
Many of the experimental observations presented in the paper require a more thorough analysis and reasoning. For example in the first paragraph on page 7, the reasons for observing the reported changes in the cohesive force should be added. The same is true about the observations regarding the internal friction angle, as reported in paragraph 2.
Author’s response: Two paragraphs are corrected to explain these experimental observations on page 7 in paragraph 1 and 2.
(1)It can be seen from Figure. 7 that the cohesive force increases firstly and then decreases. When fiber dosage was increased from 0% to 0.25%, a small amount of fiber can fill the pores between iron tailings powder, take advantage of spatial constraint effect, so as to increase the cohesive force of fiber reinforced iron tailings powder. When fiber dosage was increased from 0% to 0.25%, the agglomeration effect of fiber is increased due to the increase of fiber. The cohesive force of fiber reinforced iron tailings powder was decreased on account of the spatial constraint effect hard to proceed.
(2)With the increase of fiber dosage, the internal friction angle of polypropylene fiber reinforced iron tailings powder decreases firstly and then increases. When fiber dosage was increased from 0% to 0.25%, the fiber is dispersed among the iron tailings powder. A small amount of fiber can’t produce obvious friction effect, the internal friction angle of iron tailings is decreased due to the particle spacing of iron tailings is increased. When the fiber dosage was increased from 0.25% to 1%, the combination form between fiber and iron tailings were diversified due to the fiber density in the same volume increased, the angle of internal friction was large due to the strengthen of friction effect .
We have added this explanation to the manuscript, see lines 157 to 162 and lines 166 to 172.
Also, in the first paragraph on page 10, the relation between the normal stress, the fiber dosage, and the shear energy dissipation should be justified.
Author’s response: According to table 3 and figure 9, there is a linear relationship between direct shear energy dissipation, normal stress and fiber content. We have changed “It can be concluded from Table 3 and Figure. 10 that, under different normal stresses, the…” to “According to table 3 and figure 9, there is a linear relationship between direct shear energy dissipation, normal stress and fiber content. The… ”, see lines 236 to 237.
Other places in the manuscript have also been modified by the authors.

Reviewer 2 Report
The manuscript “Investigation on direct shear and energy dissipation characteristics of iron tailings powder reinforced by polypropylene fiber” prepared a composite of iron tailings powder and polypropylene fiber. The composites with different polypropylene fiber contents of 0%, 0.25%, 0.5%, 0.75%, and 1% were prepared. Their mechanical properties were characterized using various stress of 100, 200, 300, and 400 kPa. The results show that the polypropylene fiber can improve the shear properties of the iron tailings powder. The work is very interesting and meaningful to make full use of iron waste. This manuscript is well prepared, which could be published after solving several minor issues.
The tense through all the manuscript should keep consistent. Sometimes, the authors use the present tense, and sometimes the authors use the past tense. For example, on page 3 line 106, “The sample is formed by compaction” should be “The sample was formed by compaction”. There are some grammar errors, which should be properly corrected. For example, in page 5 line 127, “(a)-(e) is correspond to the F-s curve” should be “(a)-(e) are correspond to the F-s curve” There should be a space between numbers and units. For example, on page 6 line 143, “4mm” should be “4 mm” In Figure 9, all the conditions are expressed in the same color and style lines, which is hard to clearly see the details.Author Response
The authors sincerely thank the reviewer for taking the time and effort to review our manuscript, raise key issues and questions, and provide valuable comments and feedback.Thanks for the encouraging recommendation for the revised draft.
The tense through all the manuscript should keep consistent. Sometimes, the authors use the present tense, and sometimes the authors use the past tense. For example, on page 3 line 106, “The sample is formed by compaction” should be “The sample was formed by compaction”.
Author’s response: Authors are very sorry for this mistake. We have corrected “The sample is formed…” to “The sample was formed”, see the page 3 line 106.
Another problem was corrected from “the upper and lower surfaces of the sample are flattened” to “the upper and lower surfaces of the sample were flattened”, see the page 4 line 107.
There are some grammar errors, which should be properly corrected. For example, in page 5 line 127, “(a)-(e) is correspond to the F-s curve” should be “(a)-(e) are correspond to the F-s curve”
Author’s response: We have corrected “Figure. 7, (a)-(e) is correspond to…” to “Figure. 7, (a)-(e) are correspond to”, see the page 5 line 127.
There should be a space between numbers and units. For example, on page 6 line 143, “4mm” should be “4 mm”
Author’s response: We have corrected from “According to the F-s curve, the shear force corresponding to the 4mm” to “According to the F-s curve, the shear force corresponding to the 4 mm”, see page 6 line 143.
In Figure 9, all the conditions are expressed in the same color and style lines, which is hard to clearly see the details.
Author’s response: Authors are very sorry for this mistake, figure 7, figure 9 and figure 11 were redrew in different colors to express clear details. See lines 150, 234 and 270.
5. Other places in the manuscripthave also been modified by the authors.
